# Rac1 Temporarily Suppresses Fertilization Envelope Formation Immediately After 1-Methyladenine Stimulation

**DOI:** 10.3390/cells14060405

**Published:** 2025-03-10

**Authors:** Sakurako Aida, Takako Matsumoto, Yuna Yamazaki, Nunzia Limatola, Luigia Santella, Kazuyoshi Chiba

**Affiliations:** 1Department of Biological Sciences, Ochanomizu University, Tokyo 112-8610, Japan; sakurako.id@gmail.com (S.A.); takakom506@gmail.com (T.M.); 051124yn@gmail.com (Y.Y.); 2Department of Research Infrastructures for Marine Biological Resources, Stazione Zoologica Anton Dohrn, 80121 Napoli, Italy; nunzia.limatola@szn.it (N.L.); luigia.santella@szn.it (L.S.)

**Keywords:** starfish, oocyte maturation, Rac1, actin, calcium, fertilization, fertilization envelope

## Abstract

In starfish oocytes, the hormone 1-methyladenine (1-MA) induces germinal vesicle breakdown (GVBD) through a signaling cascade involving PI3K, SGK, Cdc25, and Cdk1/cyclin via G-proteinβγ subunit. Following GVBD, fertilization triggers an intracellular calcium increase, leading to the formation of the fertilization envelope (FE) via cortical granule exocytosis. While transient calcium elevations are known to occur after 1-MA stimulation even without fertilization, the inability of these calcium elevations to induce cortical granule exocytosis and FE formation remained unexplained. In this study, we found that co-treatment with 1-MA and calcium ionophore A23187 prevents FE formation, revealing a transient period termed the “no FE phase” persisting for several minutes. After no FE phase, the oocytes regain full competence to form the FE. Furthermore, we identified that the GEF/Rac1 signaling cascade is activated during the no FE phase. Notably, constitutively active Rac1 expressed in oocytes reproduces this inhibition even in the absence of 1-MA stimulation. These findings suggest that the GEF/Rac1 cascade, triggered by 1-MA, initiates the no FE phase and plays a critical role in coordinating the progression of subsequent fertilization events.

## 1. Introduction

Starfish oocytes are arrested at the prophase of meiosis I in the ovaries, and hormonal stimulation by 1-methyladenine (1-MA) induces germinal vesicle breakdown (GVBD) [1]. After GVBD, oocytes are spawned, and they acquire the ability to form a complete fertilization envelope (FE) in response to fertilization in the seawater or calcium ionophore A23187, which increases intracellular calcium ion levels [2,3]. Prior to hormonal stimulation, however, oocytes remain immature, displaying only partial FE elevation or vitelline envelope elevation due to insufficient inositol 1,4,5-trisphosphate (IP_3_)-mediated calcium release and inadequate calcium-induced cortical granule exocytosis [4,5]. Interestingly, oocytes lose the ability to form even a partial FE after 1-MA treatment but before GVBD [3]. Notably, simultaneous treatment with A23187 and 1-MA prevents FE formation entirely, indicating that 1-MA exerts an inhibitory effect on FE formation or exocytosis, though the precise mechanism remains unclear [3].

Starfish oocytes express a heterotrimeric G-protein, and the Gα subunit is ADP-ribosylated by pertussis toxin, which inhibits 1-MA-induced GVBD [6,7]. Because starfish Gβγ injection into oocytes induces GVBD [8,9], 1-MA stimulation likely promotes the dissociation of Gαi from Gβγ. The released Gβγ then activates PI3K [8,10], leading to the phosphorylation and activation of SGK via TORC2 and PDK1 [11,12]. In turn, SGK phosphorylates cdc25 and myt1, ultimately activating Cdk1 and triggering GVBD [11,12]. While the signaling events linking PI3K to SGK activation remain unresolved, additional unidentified signaling components appear to contribute, as neither mutant Gβγ nor PI3K alone is sufficient to induce GVBD, but their co-expression restores GVBD [13]. Furthermore, mammalian Gβγ is known to activate small G-protein guanine nucleotide exchange factors (GEFs) [14,15,16], suggesting that both GVBD and the 1-MA-induced inability to form a FE before GVBD in starfish may be regulated via GEF activation.

The actin cytoskeleton is regulated by the RHO family of a guanine nucleotide-binding protein, including RAC1, which controls the protrusion of lamellipodia [17]. In this study, we investigated whether GEFs and the small G-protein Rac1 are involved in the loss of FE-forming capacity after 1-MA treatment but prior to GVBD. Understanding this regulatory mechanism could provide insights into the block of FE formation before fertilization, as 1-MA stimulation transiently increases intracellular calcium levels [18,19], which is regulated by actin cytoskeleton in the cortex of the starfish oocytes [20,21].

## 2. Materials and Methods

### 2.1. Animals and Oocyte Preparation

The starfish *Asterina/Patiria pectinifera* is found in the shallow subtidal zones along the coasts of Japan and has been used as a model organism in developmental biology. They were collected during the breeding season and were kept in laboratory aquaria with filtered seawater at 10–16 °C. Ovaries were collected from female animals, and oocytes were squeezed from the ovaries using tweezers in the ice-cold Ca^2+^-free seawater (462 mM NaCl, 10 mM KCl, 36 mM MgCl_2_, 17 mM MgSO_4_, 10 mM EPPS, pH 8.2). To remove follicle cells, they were washed several times with ice-cold Ca^2+^-free seawater and incubated in artificial seawater (462 mM NaCl, 9 mM CaCl_2_, 10 mM KCl, 36 mM MgCl_2_, 18 mM MgSO_4_, 20 mM H_3_BO_3_, pH 8.2) at 20 °C. The oocytes having GV (called GV oocytes) were selected and used for experiments.

### 2.2. Observation of the Maturing Process and Inhibitor Treatment

The timing of germinal vesicle breakdown (GVBD) was monitored under a microscope at designated time points before (0 min) and after treatment with artificial seawater containing 1 µM 1-MA. Oocytes were pre-treated with 100 µM Rac1 inhibitor NSC23766 [22] (Abcam, Cambridge, UK) for 1 h or 20 µM PI3K inhibitor Wortmannin [23] (LC laboratories, Woburn, MA, USA) for 30 min, followed by incubation with or without 0.25 µM 1-MA. The timing of GVBD was then monitored under the same conditions.

### 2.3. Observation of Fertilization Envelope Formation

Fertilization envelope (FE) formation or vitelline envelope elevation was observed as described by Chiba and Hoshi [3]. Briefly, immature or maturing oocytes were treated with or without 20 µM calcium ionophore A23187 [24] (Sigma-Aldrich, St. Louis, MO, USA), which can activate echinoderm eggs [25]. The FE formation and its morphology were classified into three categories: partial FE (abnormal or incomplete FE), no FE (invisible FE), and complete FE (normal and fully formed FE) (Appendix A: In general, complete FE was observed in germinal vesicle breakdown (GVBD) oocytes treated with A23187. In contrast, partial FE (abnormal FE) was observed in immature oocytes treated with A23187 in the absence of 1-MA. FE formation was not observed (no FE) in oocytes treated with A23187 after 1-MA treatment but before GVBD [3]). Images of FE formation were captured using a DIC microscope and a camera system.

### 2.4. Sample Preparation and SDS-PAGE

Oocytes were recovered in 5 µL of seawater, mixed with 5 µL of sample buffer (125 mM Tris-HCl pH 6.8, 20% glycerol, 4% SDS, 10% 2-mercaptoethanol, 0.01% bromophenol blue), and frozen in liquid nitrogen. After thawing and boiling at 95 °C for 5 min, proteins in the sample buffer were separated by SDS-PAGE using gels with different concentrations: 8.5% for Cdc25, SGK, Akt, and Cdk1; 12% for Cdk1 and Rac1; and 13% for Rac1.

### 2.5. Western Blotting

Proteins were transferred to PVDF membranes (Merck, Darmstadt, Germany) using a semi-dry blotting system. Membranes were blocked and incubated with the following primary antibodies at specified dilutions: Anti-sfCdc25 (1:1000, Can Get Signal Immunoreaction Enhancer Solution 1, TOYOBO, Tokyo, Japan) [26]; Anti-Cdc25-pS188 (1:1000, Can Get sol. 1, TOYOBO, Tokyo, Japan) [13]; Anti-SGK-pAloop (1:50, in TBS-T) [12]; Anti-sfSGK-HM (1:1000, in TBS-T) [11]; Anti-sfAkt phospho-Ser477 (1:1000) [27]; Anti-sfAkt C-terminal 88 amino acids fragment (1:1000, Can Get sol. 1) [28]; Anti-Cdk1 phospho-Tyr15 (1:1000, Can Get sol. 1); Anti-Cdk1 (PSTAIR) (1:1000, Can Get sol. 1); Anti-mouse Rac1 (clone 23A8, Sigma-Aldrich, St. Louis, MO, USA) (1:1000, in TBS-T); Anti-Myc tag (Cat# ab9106, Abcam, Cambridge, UK) (1:1000, Can Get sol. 1). HRP-conjugated secondary antibodies (anti-rabbit IgG or anti-mouse IgG) were used at a 1:2000 dilution in TBS-T or Can Get sol. 2. Proteins reactive with the antibodies were detected using the ECL Prime Western Blotting Detection System (GE Healthcare, Chicago, IL, USA). Digital images were captured with the LAS-4000mini Luminescent Image Analyzer (FUJIFILM Wako Pure Chemical, Osaka, Japan).

### 2.6. cDNA Cloning of Rac1

To obtain a cDNA encoding starfish homologue of *Rac1*, total mRNA was isolated from starfish immature oocytes using TRI regent (Molecular Research Center, Cincinnati, OH, USA). The first-strand cDNA library was prepared from total mRNA using the SMARTerTM RACE cDNA Amplification Kit (Clontech, Mountain View, CA, USA). PCR was performed using a starfish Rac1 gene-specific 3′ primer (TGCAGCGGTATCAGAGTGTG) and a 5′ primer (GCGTGTTGGCTATTGCACTT) with an annealing temperature of 55 °C, resulting in the amplification of an approximately 600 bp product. PCR products were purified by agarose gel electrophoresis and extracted from the gel using the Wizard SV Gel and PCR Clean-Up System (Promega, Madison, WI, USA). Purified PCR products were cloned into a pCR2.1-TOPO vector (Invitrogen, Carlsbad, CA, USA), and insert sequences were determined using Sanger sequencing with the Applied Biosystems™ 3130 DNA Analyzers (Applied Biosystems, Waltham, MA, USA). The GenBank accession number of sf Rac1 is LC848442.

### 2.7. DNA Constructs

To express wild *Rac1* in starfish oocytes, starfish *Rac1* was amplified by PCR using primers which had a myc-tag sequence (forward: 5′-CAGAAGCTGATCTCAGAGGAGGACCTGATGCAAGCCATCAAATGTGTCG-3′, and reverse: 5′-GTGGTAACCAGATCTTTATATCAAGCTGCATTTGGGCC-3′), followed by PCR using primers for In-Fusion (forward: 5′-ACCGAATTCTACAATATGGAGCAGAAGCTGATCTCAGAGGAGG-3′, and reverse: 5′-GTGGTAACCAGATCTTTATATCAAGCTGCATTTGGGCC-3′), with an annealing temperature of 55 °C, resulting in the amplification of an approximately 600 bp product. The psfSGK vector including *A. pectinifera cyclin B* Kozak sequence (5′-TACAAT-9′) [11] was amplified by PCR using primers (forward: 5′-AGATCTGGTTACCACTAAACCAGCCTCAAG-3′, and reverse: 5′-ATTGTAGAATTCGGTACCGATCTGCCAAAG-3′), with an annealing temperature of 55 °C, resulting in the amplification of an approximately 3000 bp product. These fragments were ligated using the In-Fusion HD Cloning Kit (Clontech, Mountain View, CA, USA), and the resultant plasmid was named psfRac1_WT.

To express constitutively active *Rac1* (Q61L) in starfish oocytes, psfRac1_WT was amplified using primers for mutagenesis (forward: 5′-GCGGGACTGGAGGACTACGACAGACT-3′, and reverse: 5′-GTCCTCCAGTCCCGCCGTATCCCACA-3′), with an annealing temperature of 55 °C, resulting in the amplification of an approximately 4000 bp product, and the PrimeSTAR^®^ Mutagenesis Basal Kit (Takara Bio, Shiga, Japan), and the resultant plasmid was named psfRac1_CA.

To express fluorescence resonance energy transfer (FRET) sensor visualizing an activity of Rac1 in starfish oocytes, the *RaichuEV-Rac1* sequence was amplified using *pCAGGS-RaichuEV-Rac1* [29] and specific primers (forward: 5′-ACCGAATTCTACAATGCTGGTTGTTGTGCTGTCTC-3′, and reverse: 5′-TTCTTGAGGCTGGTTGTCAGATGCTCAAG GGCTT-3′), followed by ligation with the *psfSGK* vector amplified using primers (forward: 5′-AACCAGCCTCAAGAACACCCG-3′, and reverse: 5′-ATTGTAGAATTCGGTACCGATCTGCC-3′) and the In-Fusion HD Cloning Kit (Clontech, Mountain View, CA, USA). The resultant plasmid was named psfRaichuEV-Rac1.

The in vitro transcription of wild *Rac1*(*psfRac1_WT*), mutant *Rac1* (*psfRac1_CA*) and *RaichuEV-Rac1* was performed using the mMESSAGE mMACHINE T7 kit (Ambion Thermo Fisher Scientific, Waltham, MA, USA) and a poly A tailing kit (Ambion Thermo Fisher Scientific, Waltham, MA, USA). They were purified using phenol/chloroform extraction and ethanol precipitation.

### 2.8. Microinjection

Microinjection was performed as previously described [6]. Briefly, the oocytes were injected using a constricting pipet filled with in vitro synthesized RNAs (10 pg/oocyte), Suc1 protein (1 ng/oocyte), or anti-SGK antibody (200 pg/oocyte). Injected oocytes were incubated for the indicated periods.

### 2.9. Rac1 Pull Down

Active Rac1 of starfish was pulled down using a Rac activation kit Cat#17-283 (Merck Millipore, Darmstadt, Germany): Sedimented 1000 oocytes were frozen in liquid nitrogen, dissolved in 50 µL Mg^2+^ Lysis/Wash Buffer (25 mM HEPES pH 7.5, 150 mM NaCl, 5% lgepal CA-630, 50 mM MgCl_2_, 5 mM EDTA, 10% glycerol, 10% artificial seawater, protease inhibitor cocktail; Merck, Darmstadt, Germany), and then mixed occasionally for 30 min on ice. The supernatant (oocyte lysate) was recovered by centrifugation (14,000 rpm, 15 min). PAK1 agarose beads to pull down Rac1 were incubated in the oocyte lysate with GTPγS or GDP. PAK1 agarose beads were washed with Mg^2+^ Lysis/Wash Buffer three times and boiled in sample buffer for SDS-PAGE to release Rac1 from the beads. Rac1 proteins were transferred to PVDF membranes (Merck, Darmstadt, Germany) using a semi-dry blotting system. Rac1 on the membrane was detected using Anti-mouse Rac1 (clone 23A8, Sigma-Aldrich, St. Louis, MO, USA) (1:1000, in TBS-T).

### 2.10. FRET Measurement Using Confocal Microscopy

The RaichuEV-Rac1 biosensor, which consists of the donor CFP and acceptor YFP, was expressed in starfish oocytes by a microinjection of the mRNA. The excitation of CFP was performed at a wavelength of 405 nm, and emitted fluorescence from CFP (415–511 nm) and YFP (514–604 nm) was detected using a laser scanning confocal microscope (LSM710, Carl Zeiss, Germany) with a scan time of 3.87 s. Given that the molar ratio of CFP to YFP in the RaichuEV-Rac1 biosensor remains constant at every pixel, the FRET efficiency was calculated as the simple ratio between the fluorescence intensities of CFP and YFP (YFP intensity/CFP intensity) [30]. To calculate the average fluorescence intensities, regions of interest (ROI) encompassing individual oocytes were determined. CFP and YFP fluorescence intensities were measured during 1-MA treatment, exported in Excel format, and processed to generate line graphs. To visualize FRET images during the time course of 1-MA treatment, pixel-based calculations of the FRET efficiency (YFP-to-CFP ratio) were performed using Carl Zeiss software ZEN.

## 3. Results and Discussion

### 3.1. Changes in FE Formation upon 1-MA Stimulation and the Involvement of Rac1 in the No FE Phase

It has been established that co-treatment with 1-MA and A23187 prevents fertilization envelope (FE) formation and that the full ability to form an FE is only acquired after GVBD [3]. However, the precise timing of the recovery of FE-forming capacity following 1-MA treatment remained unclear. To address this, oocytes were treated with A23187 for 10 min at 3, 6, 9, and 12 min after 1-MA exposure. The results revealed that the loss of FE-forming ability persisted until 6 min after 1-MA treatment (Figure 1A,B). We define this period as the “no FE phase”. By 9 min, the no FE phase had ended, and the oocytes regained the ability to form a complete FE. In this batch, GVBD occurred at 18 min, indicating that the no FE phase concludes at approximately half the time required for GVBD. Notably, the dephosphorylation of Cdk1 was observed at the end of the no FE phase (Figure 1A,C), suggesting that Cdk1 activation may be involved in the termination of this phase. Similarly, oocytes from other individuals exhibited no FE phases that ended at about half the time to GVBD, coinciding with the activation of Cdk1 (Appendix A). A microinjection of MPF containing active Cdk1 solely induces GVBD, and a complete FE is formed upon fertilization [7], supporting the theory that the no FE phase is canceled by Cdk1.

To identify the molecular components of the 1-MA signaling pathway involved in the initiation of the no FE phase, we injected Suc1, an inhibitor of Cdk1/cyclin B [31,32], into immature oocytes. Following A23187 treatment, these oocytes formed partial FEs (Figure 1D,F). Similarly, partial FE formation was observed when oocytes were injected with a functional-blocking antibody against SGK or treated with a PI3K inhibitor (Figure 1D,G,H). These findings indicate that the signaling cascade downstream of PI3K does not contribute to the initiation of the no FE phase.

Next, we tested the effect of NSC23766, an inhibitor of guanine-nucleotide exchange factor (GEF)/Rac1 [22], which is potentially activated by Gβγ. NSC23766 treatment allowed formation of the partial FE to occur with simultaneous application of 1-MA and A23187 (Figure 1D,E). Furthermore, NSC23766 not only inhibited 1-MA-induced GVBD (Figure 1I) but also blocked the activation of SGK, Akt, Cdc25, and Cdk1/cyclin B (Figure 1J). These findings indicate that Gβγ activates both the PI3K and GEF/Rac1 signaling pathways, with the latter playing a pivotal role in initiating the no FE phase. It is also possible that GEF/Rac1 corresponds to the signaling cascade that could not be activated by mutant Gβγ [13].

### 3.2. Presence and Activity of Rac1 Protein in Starfish Oocytes

To confirm the expression of Rac1 in starfish oocytes, we performed Western blot analysis using an anti-mammalian Rac1 antibody, which detected a 21-kDa band (Figure 2A). Since active mammalian Rac1 is known to interact with PAK1 [33], we conducted a pull-down assay with mammalian PAK1-conjugated beads and lysates from starfish oocytes in the presence of GTPγS to activate Rac1. A 21-kDa band was observed in samples treated with GTPγS (Figure 2B, lane 1) but not in samples treated with GDP, which maintains Rac1 in its inactive state (Figure 2B, lane 2). These findings confirm that the 21-kDa band corresponds to starfish Rac1.

Furthermore, the expression of a FRET sensor for Rac1 [34] within oocytes revealed that 1-MA stimulation led to an increase in FRET signal (Figure 2C, Appendix A), confirming that Rac1 is not only present in starfish oocytes but is also activated in a 1-MA-dependent manner.

### 3.3. Constitutively Active Rac1 Induces the No FE Phase

If Rac1 is truly involved in the no FE phase, the expression of active Rac1 within oocytes should induce this phase. To test this, we cloned the cDNA of starfish *Rac1* and generated a constitutively active *Rac1* mutant. The amino acid sequence of the cloned sfRac1 showed a high degree of similarity to that of human Rac1 (Figure 3A). Notably, the amino acid sequence at the GEF interaction site of sfRac1 was identical to that of human Rac1. As expected, the expression of the constitutively active *Rac1* triggered the no FE phase independently of 1-MA stimulation (Figure 3B,C). Importantly, this constitutively active Rac1 did not inhibit 1-MA-dependent GVBD, confirming that it does not interfere with the 1-MA signaling pathway (Figure 3D). These results strongly suggest that the inhibition of FE formation was not caused by unintended toxicity from constitutively active Rac1 but rather by its specific involvement in the no FE phase. Notably, the expression of Rac1 alone did not induce GVBD, nor did it activate the molecules involved in GVBD (Figure 3E). A dominant negative Rac1 [35] may be useful for further studies.

How Rac1 induces the no FE phase remains to be determined, but its known role in actin polymerization suggests that actin may be involved. Rac1 promotes the formation of branched actin filament networks through the WAVE complex, contributing to the formation of lamellipodia [36]. Indeed, it is well established that the structure and dynamics of the cortical F-actin of immature oocytes are dramatically reorganized following 1-MA stimulation to make the mature oocyte successfully fertilizable [19,21,37,38].

The physiological significance of the no FE phase may lie in preventing “erroneous” FE formation triggered by the transient calcium increase induced by 1-MA. Physiological FE formation is normally induced by intracellular calcium elevation following fertilization, which occurs after GVBD. Therefore, a mechanism that prevents premature FE formation in response to earlier calcium increases is essential. Our findings demonstrate that 1-MA-dependent Rac1 activation ensures that FE formation does not occur in response to the transient calcium rise induced by 1-MA.

Although it has not been studied whether Rac1 is involved in FE formation, Rac1 exhibits upregulated expression during infection in sea urchins [39] and sea cucumbers [40]. In addition, mammalian Rac1 is involved in the acrosome reaction [41] and sperm motility [42]. Future studies elucidating how the Rac1 and/or GEF pathway contributes to SGK activation may provide further insights into the mechanisms underlying the resumption of meiosis.

## Figures and Tables

**Figure 1 cells-14-00405-f001:**
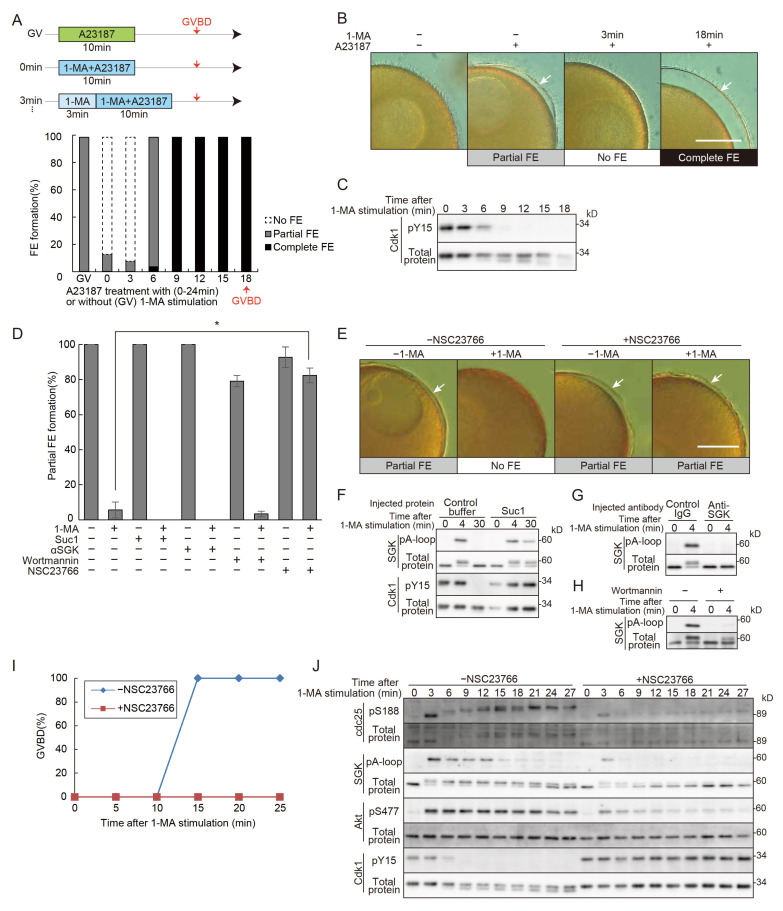
Changes in FE formation upon 1-MA stimulation and the involvement of Rac1 in the no FE phase. (**A**) The upper panel illustrates the experimental schedule, while the lower panel presents the results. In the upper panel: GV, untreated condition; 0 min, stimulation with both 1-MA and A23187 for 10 min; 3 min, treatment with 1-MA alone for 3 min, followed by the addition of A23187 and continued stimulation with both reagents for 10 min. In the lower panel: The proportion of partial or absent FE (no FE) formation is shown for the GV, 0 min, and 3 min conditions. Additionally, treatments at 6, 9, 12, 15, and 18 min involved stimulation with 1-MA alone for the specified durations, followed by the additional A23187 plus 1-MA treatments for 10 min. The proportion of partial or complete FE formation is shown. (**B**) Representative images of fertilization envelopes observed in the experiments described in (**A**). Scale bar: 100 μm. Arrows indicate partial or complete FE. (**C**) In the same oocytes used in (**A**), samples were collected every 3 min following 1-MA stimulation, and Cdk1 dephosphorylation was monitored by Western blotting. The no FE phase terminated in parallel with the onset of Cdk1 dephosphorylation, showing Cdk1 activation. (**D**) Fertilization envelope formation rates were quantified following a microinjection of Suc1 protein or anti-SGK inhibitory antibodies into oocytes, or after treatment with Wortmannin or NSC23766, followed by co-stimulation with 1-MA and A23187. While the no FE phase was maintained in the Suc1, anti-SGK, and Wortmannin treatments, oocytes treated with 100 μM NSC23766 displayed a higher proportion of partial FE formation, with the no FE phase abolished compared to controls. Error bars indicate the standard error of the mean for three independent experiments. A Tukey HSD test (* *p* < 0.05). (**E**) Representative images of fertilization envelopes observed in the NSC23766 treatment described in (**D**). Scale bar: 100 μm. Arrows indicate partial FEs. (**F**) Cdk1 inactivation following Suc1 treatment was monitored by Western blotting. (**G**) SGK inhibition in oocytes injected with anti-SGK antibodies was monitored by Western blotting. (**H**) SGK inhibition by Wortmannin treatment was monitored by Western blotting. (**I**) GVBD rates were plotted following 1-MA stimulation in NSC23766-treated oocytes. Treatment with 100 μM NSC23766 inhibited GVBD. (**J**) Western blot analysis was performed on samples collected every 3 min following 1-MA stimulation in NSC23766-treated oocytes. Although SGK was partially phosphorylated at 3 min, subsequent dephosphorylation was observed, with the complete dephosphorylation of Cdk1 not occurring.

**Figure 2 cells-14-00405-f002:**
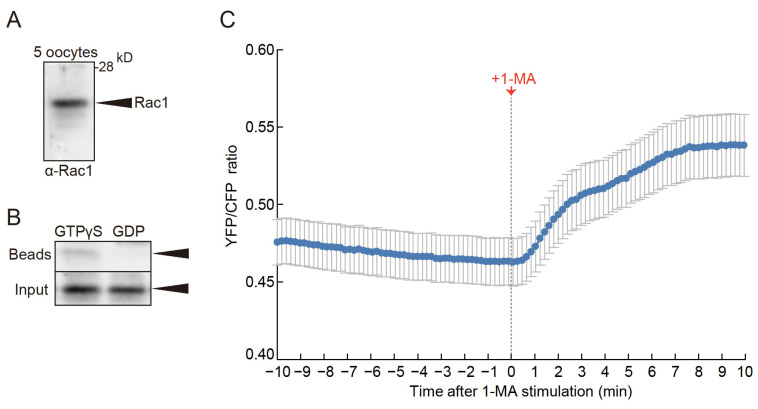
Presence and activity of Rac1 protein in starfish oocytes. (**A**) Western blotting with anti-mammalian Rac1 antibodies confirmed the presence of Rac1 in oocytes, with a band detected near the expected molecular weight of 21 kDa. (**B**) Lysates from immature oocytes were treated with either GTPγS (lane 1) or GDP (lane 2) and subjected to pulldown assays using beads conjugated with mammalian PAK1. The GTPγS-treated lysates showed stronger bands, while GDP-treated lysates displayed weaker bands, indicating that Rac1 from starfish oocytes interacts with mammalian PAK1. Arrows indicate Rac1 bands. (**C**) Using a FRET sensor, endogenous Rac1 activity during oocyte maturation was monitored in real-time. Upon 1-MA stimulation, Rac1 activity increased. Error bars indicate the standard error of the mean for ten independent experiments.

**Figure 3 cells-14-00405-f003:**
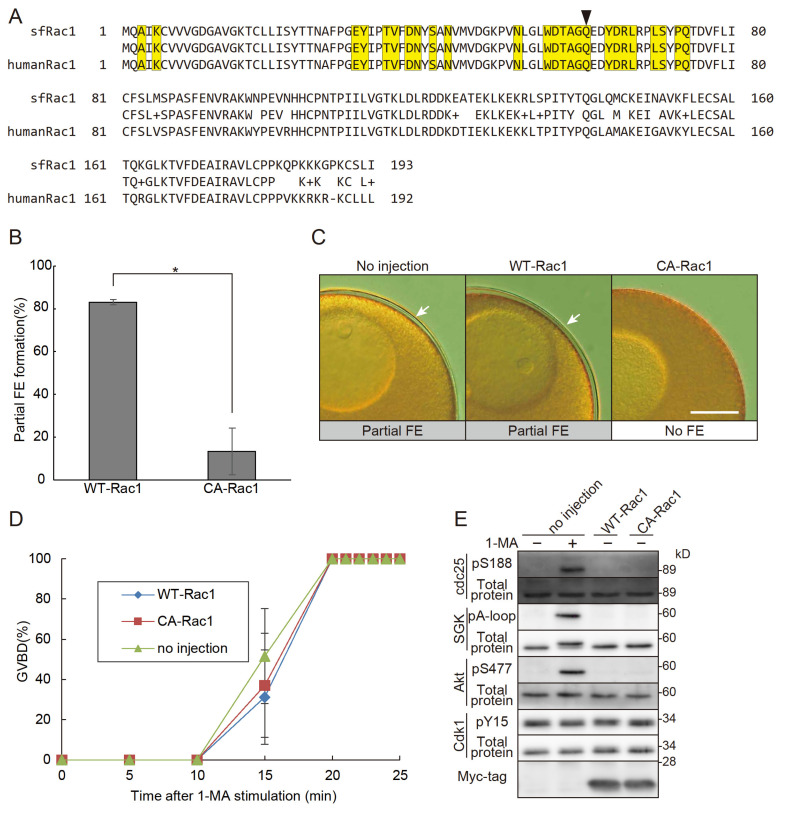
Constitutively active Rac1 induces the no FE phase. (**A**) The amino acid sequence of sfRac1 was aligned with that of human Rac1 (Sequence ID: NP_008839.2). The middle column represents the quality of the alignment: identical amino acids are shown as letters, “+” indicates amino acids with similar characteristics, and dashes represent gaps. Yellow boxes highlight the GEF interaction site. The arrow marks the glutamine residue replaced by leucine to generate constitutively active sfRac1 (Q61L). (**B**) Oocytes expressing constitutively active Rac1 (CA-Rac1) were treated with A23187, and the rates of partial FE formation were assessed. Oocytes expressing CA-Rac1 exhibited no FE. Error bars indicate the standard error of the mean for three independent experiments. Student’s *t*-test (* *p* < 0.05). (**C**) Representative images of fertilization envelopes from the CA-Rac1 experiments in (**B**). Scale bar: 100 μm. Arrows indicate partial FEs. (**D**) GVBD rates were plotted following 1-MA stimulation in CA-Rac1-expressing oocytes. The expression of CA-Rac1 did not induce spontaneous GVBD, nor did it accelerate the timing of GVBD. Error bars indicate the standard error of the mean for three independent experiments. (**E**) Western blot analysis was performed on CA-Rac1-expressing oocytes. The expression of CA-Rac1 did not alter the phosphorylation status of SGK or Cdk1.

## Data Availability

GenBank accession numbers of sf Rac1 is LC848442.

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
