# Peer review of "Rac1 Temporarily Suppresses Fertilization Envelope Formation Immediately After 1-Methyladenine Stimulation"

_cells, 2025, doi:10.3390/cells14060405_

Round 1

Reviewer 1 Report

Comments and Suggestions for Authors

As attach 

Author Response

We would like to thank the reviewer for constructive comments and suggestions.

Comments 1: There is insufficient information about starfish, particularly the study species Asterina pectinifera. I suggest adding a dedicated paragraph on this species, covering its ecology, salt tolerance levels, and the rationale for selecting it for the study.

Response 1: As you suggested, we added a paragraph on the starfish in Materials and Methods:

(Line 62-63) The starfish Asterina/Patiria pectinifera is found in the shallow subtidal zones along the coasts of Japan and has been used as a model organism in developmental biology.

Comments 2: Similarly, provide a brief introduction to Rac1 to capture the reader's attention. Currently, Rac1 is not mentioned in the introduction, making its sudden appearance in the objective about FE formation feel abrupt. Try to establish a smoother connection between the objective and the introduction

Response 2: Thank you for your comments. We added a brief introduction to Rac1 as follows:

(Line 53-54) The actin cytoskeleton is regulated by the RHO family of a guanine nucleotide-binding protein, including RAC1, which controls the protrusion of lamellipodia.

Comments 3: Here’s a refined version of your sentence for better clarity and correctness:

I am afraid that I cannot find your refined version of the sentence in the HP of “Reply to Reviewers”, probably due to accidental deletion of your comment.

Comments 4: How many starfish samples were taken for the study, and what was the average weight of the samples?

Response 4: In this study, we used 18 animals (100-300g/animal).

Comments 5: What is the amplicon size of the Rac1 gene used for PCR? It would be helpful to present this information in a table, possibly as a supplementary table, listing the primers and genes used along with details such as annealing temperature, amplicon size, and other relevant parameters

Response 5: As indicated, we added each annealing temperature and amplicon size in Materials and Methods (Lines 119-120, 134-135, 138-139, 144-146.)

Comments 6: Line 83: I see that this is a relatively new work. It would be beneficial to include a clear image illustrating the three categories mentioned: Partial FE, No FE, and Complete FE. You may add this image in the Methods section or as a supplementary figure; I leave the placement up to you.

Response 6: As indicated, we added the Figure S1 showing the three phases.

(Line 85-89) Fig. S1: In general, complete FE was observed in germinal vesicle breakdown (GVBD) oocytes treated with A23187. In contrast, partial FE (abnormal FE) was observed in immature oocytes treated with A23187 in the absence of 1-MA. FE formation was not observed (no FE) in oocytes treated with A23187 after 1-MA treatment but before GVBD [3].

(Line 334) Figure S1: Illustration of the three categories of FE: Partial FE, No FE, and Complete FE [3].

Comments 7: Line 122. You already have Sanger seq, and accession no Write the number of Rac1 here in this line (LC848442.)

Response 7: As indicated, we added a paragraph in Materials and Methods:

(Line 125-126) GenBank accession numbers of sf Rac1 is LC848442.

Comments 8: I didn’t find any statistical analysis indicating significance (mainly in Fig 1 and 3). What type of test did you use to determine statistical significance?

Response 8: In the previous version of our manuscript, we indicated statistical significance in the figure legends. In the new version, (Line248-259) “A Tukey HSD test (p < 0.05)”, and (Line325-326) “Student's t-test (p<0.05)”.  Please check figure legends of Fig.1 and Fig.3 of the old version.

Comments 9: The supplementary video is not clear to me. Would it be possible to use an arrow or a circle within the video to highlight what you are trying to indicate?

Response 9: Thank you for your comments. As you indicated, it may be difficult to observe the gradual color changes in the oocyte. To address this, we have added the following explanation:

(Line 344-346) The oocyte was treated with 1-MA 10 minutes after the start of the movie. Following 1-MA stimulation, a gradual color change from blue to blue-green, indicating an increase in Rac1 activity, was observed.

Reviewer 2 Report

Comments and Suggestions for Authors

Rac1 Temporarily Suppresses Fertilization Envelope Formation Immediately after 1-Methyladenine Stimulation

Overview

This study investigates the role of Rac1 in regulating fertilization envelope (FE) formation in starfish oocytes following 1-methyladenine (1-MA) stimulation. The authors identify a transient inhibitory phase, termed the "no FE phase," during which Rac1 activation prevents premature FE formation before germinal vesicle breakdown (GVBD). Using various tests, they provide evidence that Rac1 plays a crucial role in modulating oocyte maturation and fertilization competency.

The study introduces the concept of the "no FE phase," a previously unrecognized transient period where oocytes lose the ability to form an FE after 1-MA stimulation. The work suggests that Rac1 activation, triggered by GEFs in response to 1-MA, is responsible for this inhibition. This finding is significant as it provides insight into how premature FE formation is prevented, ensuring proper fertilization timing, but certain aspects require further clarification and contextualization:

  • Could Rac1 activation be a general mechanism in other marine species, or is it specific to starfish? Do the authors expect similar regulatory pathways to function in mammals or other invertebrates? A couple of lines in the discussion would be helpful.

  • Were the Western blots quantified to show statistical significance, or were they only visually assessed?

  • Could disrupting the "no FE phase" lead to fertilization defects, such as polyspermy?

  • Are there additional factors besides Rac1 that help regulate the transition from the "no FE phase"? A thorough literature review would enhance the robustness of the present study.

  • Could the authors verify Rac1 inhibition specificity using an independent method, such as a dominant-negative Rac1 mutant? The addition of future perspectives/works on this particular topic in your paper would be useful.

  • Did the study examine actin polymerization levels directly using phalloidin staining or other imaging techniques?

  • A graphical summary (timeline) of the signaling events would improve clarity for readers.

Author Response

We would like to thank the reviewer for constructive comments and suggestions.

Comments 1: Could Rac1 activation be a general mechanism in other marine species, or is it specific to starfish? Do the authors expect similar regulatory pathways to function in mammals or other invertebrates? A couple of lines in the discussion would be helpful.

Response 1: As indicated, we added a paragraph:

(Line 310-315) Although it has not been studied whether Rac1 is involved in FE formation, Rac1 exhibits upregulated expression during infection in sea urchins [39] and sea cucumbers [40]. In addition, mammalian Rac1 is involved in the acrosome reaction [41] and sperm motility [42]. Future studies elucidating how the GEF pathway contributes to SGK activation may provide further insights into the mechanisms underlying the resumption of meiosis.

Comments 2: Were the Western blots quantified to show statistical significance, or were they only visually assessed?

Response 2: Western blots were visually assessed.

Comments 3: Could disrupting the "no FE phase" lead to fertilization defects, such as polyspermy?

Response 3: We confirmed that insemination of oocytes during the no FE phase caused polyspermy.

Comments 4: Are there additional factors besides Rac1 that help regulate the transition from the "no FE phase"? A thorough literature review would enhance the robustness of the present study.

Response 4: As indicated, we added the following discussion:

(Line 210-211) Microinjection of MPF containing active Cdk1 solely induces GVBD, and a complete FE is formed upon fertilization [7], supporting that the no FE phase is canceled by Cdk1

Comments 5: Could the authors verify Rac1 inhibition specificity using an independent method, such as a dominant-negative Rac1 mutant? The addition of future perspectives/works on this particular topic in your paper would be useful.

Response 5: Thank you very much for your comments. As you suggested, a dominant-negative Rac1 may be useful for our future study. We added the following discussion:

(Line296) A dominant negative Rac1 [35] may be useful for further studies.

Comments 6: Did the study examine actin polymerization levels directly using phalloidin staining or other imaging techniques?

Response 6: Thank you very much for your comments. We are going to study actin polymerization.

Comments 7: A graphical summary (timeline) of the signaling events would improve clarity for readers.

Response 7: We have submitted a graphical summary to improve clarity for readers.

Reviewer 3 Report

Comments and Suggestions for Authors

The inability of calcium elevations to induce cortical granule exocytosis and fertilization envelope (FE) remained unexplained. The authors found that co-treatment with 1-MA and calcium ionophore A23187 prevents FE formation, revealing a transient period termed the “no FE phase”. Constitutively active Rac1 expressed in oocytes reproduces this inhibition. They concluded that the GEF/Rac1 cascade, triggered by 1-MA, initiates the no FE phase and plays a critical role in coordinating the progression of subsequent fertilization events.

The issues need to be discussed:

Fig1, discuss if the authors should add one more group, with only 1-Methyladenine stimulation.

Give more information about the specificity of A23187.

In Fig 3B, oocytes expressing constitutively active Rac1 (CA-Rac1) were treated with A23187. What about oocytes expressing constitutively active Rac1 (CA-Rac1) without A23187?

Comments on the Quality of English Language

No

Author Response

We would like to thank the reviewer for constructive comments and suggestions.

Comments 1: Fig1, discuss if the authors should add one more group, with only 1-Methyladenine stimulation.

Response 1: Sole 1-MA treatment did not cause FE formation as shown in Fig.1 E., the second panel.

Comments 2: Give more information about the specificity of A23187.

Response 2: A23187 was used to induce FE formation by Steinhardt and Epel (1974), and since then, it has been used by many investigators to increase intracellular calcium ion concentrations. We have added this information (Line 82). Chiba and Hoshi (1989) also used A23187 and obtained similar results with calcium buffer injection, indicating specific effects of A23187 on calcium elevation.

Comments 3: In Fig 3B, oocytes expressing constitutively active Rac1 (CA-Rac1) were treated with A23187. What about oocytes expressing constitutively active Rac1 (CA-Rac1) without A23187?

Response 3: Sole CA-Rac1 expression did not induce FE formation. We had the same images as shown in Fig.3C, right panel,

Round 2

Reviewer 3 Report

Comments and Suggestions for Authors

No further comments